# Past, Present and Future of Oncolytic Reovirus

**DOI:** 10.3390/cancers12113219

**Published:** 2020-10-31

**Authors:** Louise Müller, Robert Berkeley, Tyler Barr, Elizabeth Ilett, Fiona Errington-Mais

**Affiliations:** Leeds Institute of Medical Research (LIMR), University of Leeds, Leeds LS9 7TF, UK; lme.muller@outlook.com (L.M.); raberkeley@outlook.com (R.B.); um17tkb@leeds.ac.uk (T.B.); e.ilett@leeds.ac.uk (E.I.)

**Keywords:** reovirus, oncolytic virus, immunotherapy

## Abstract

**Simple Summary:**

Within this review article the authors provide an unbiased review of the oncolytic virus, reovirus, clinically formulated as pelareorep. In particular, the authors summarise what is known about the molecular and cellular requirements for reovirus oncolysis and provide a comprehensive summary of reovirus-induced anti-tumour immune responses. Importantly, the review also outlines the progress made towards more efficacious combination therapies and their evaluation in clinical trials. The limitations and challenges that remain to harness the full potential of reovirus are also discussed.

**Abstract:**

Oncolytic virotherapy (OVT) has received significant attention in recent years, especially since the approval of talimogene Laherparepvec (T-VEC) in 2015 by the Food and Drug administration (FDA). Mechanistic studies of oncolytic viruses (OVs) have revealed that most, if not all, OVs induce direct oncolysis and stimulate innate and adaptive anti-tumour immunity. With the advancement of tumour modelling, allowing characterisation of the effects of tumour microenvironment (TME) components and identification of the cellular mechanisms required for cell death (both direct oncolysis and anti-tumour immune responses), it is clear that a “one size fits all” approach is not applicable to all OVs, or indeed the same OV across different tumour types and disease locations. This article will provide an unbiased review of oncolytic reovirus (clinically formulated as pelareorep), including the molecular and cellular requirements for reovirus oncolysis and anti-tumour immunity, reports of pre-clinical efficacy and its overall clinical trajectory. Moreover, as it is now abundantly clear that the true potential of all OVs, including reovirus, will only be reached upon the development of synergistic combination strategies, reovirus combination therapeutics will be discussed, including the limitations and challenges that remain to harness the full potential of this promising therapeutic agent.

## 1. Oncolytic Virotherapy (OVT)

Advancements in virology and molecular biology techniques over recent decades have allowed us to exploit the anti-tumour potential of oncolytic viruses (OVs) [1]. The unique ability of OVs to exploit oncogenic signalling pathways provides a significant advantage over traditional treatment modalities. OVs are specifically defined as viruses which: (i) preferentially infect and kill malignant cells through viral replication and oncolysis, and (ii) engage the immune system to promote anti-tumour immunity. Additional mechanisms of action have also been reported, including disruption of tumour-associated vasculature or stroma and modulation of the tumour microenvironment (TME) [2,3,4].

An array of OVs—naturally occurring, attenuated, and genetically modified—have been investigated in pre-clinical models and clinical trials but only two have received approval for clinical use: (i) a genetically engineered adenovirus H101, approved in China in 2005 [5], and (ii) the Food and Drug Administration (FDA)-approved talimogene laherparepvec (T-VEC)—a herpes simplex virus type 1 (HSV-1) genetically engineered to limit neurovirulence and promote an immunostimulatory environment [6,7]. This review will provide an overview of what we have learnt about oncolytic mammalian orthoreovirus since its rise as a clinically applicable agent, we will discuss areas of active pre-clinical and clinical research and consider the challenges that exist to harness its full therapeutic potential.

## 2. The Emergence of Reovirus as a Therapeutic Agent

The Reoviridae family of viruses has found hosts in mammals, fish, birds and plants [8,9]. Three serotypes of mammalian orthoreovirus have been identified: type one Lang, type two Jones, and type three Abney and Dearing [10]. Each differs in its in vivo tropism, despite a high degree of genetic similarity [11]. Type-specific diversity occurs in the S1 gene, encoding the outer capsid σ1 attachment protein, which has undergone significant evolutionary divergence [12]. Orthoreovirus type two *Jones* was the first serotype observed to replicate specifically in malignant cell lines [13]; however, it is the mammalian orthoreovirus type three Dearing strain (T3D)—now manufactured as pelareorep but previously known as Reolysin^®^—that has made progress as a therapeutic agent. Mammalian orthoreovirus T3D (hereafter referred to as reovirus) is typically isolated from human gastrointestinal and upper respiratory tracts [14,15]. In most individuals, infection proceeds asymptomatically causing mild enteric or respiratory illness in young children and being relatively non-pathogenic in adults, in line with its designation as a respiratory enteric orphan virus (reovirus) [10]. There have been sporadic reports of severe pathology associated with reovirus infection in infants and immunocompromised individuals [9,16,17,18,19,20,21] and more recently, reovirus has been implicated in coeliac disease by promoting a T_H_1 immune response, a response that bodes well for its use as an immunotherapeutic tool although oral delivery should be avoided to limit these potential unwanted side effects [22].

Reovirus is a non-enveloped, double-stranded (ds) RNA virus approximately 85 nm in diameter, with two concentric icosahedral protein capsids [23]. The outer and inner capsids protect the dsRNA genome which comprises 23.5 kbp in ten segments termed large (L1-3), medium (M1-3), or small (S1-4) according to size [23,24,25]. The gene segments encode eight structural proteins (λ1-3, µ1-2, and σ1-3) and the non-structural proteins, µNS and σNS [26]. μ1 and σ3 form part of the outer capsid, λ3 forms a subunit of the RNA polymerase and σ1 and λ2 are important for viral attachment, although σ1 initiates target cell entry [23]. The proteins also protect the virus from immune-surveillance by preventing a host anti-viral interferon (IFN) response; σ3 binds to dsRNA and prevents its binding to dsRNA-dependent protein kinase R (PKR; a dsRNA sensor) [27] and μNS sequesters the IFN transcription factor (interferon regulatory factor 3; IRF3) and inhibits its translocation to the nucleus [28].

## 3. Tumour Specificity and Replication

The reovirus life-cycle is shown in Figure 1. Viral entry occurs over multiple steps, the first being a low-affinity “tethering” of the reovirus σ1 protein to cell surface sialic acid [29,30]. Subsequently, σ1 engages junctional adhesion molecule A (JAM-A), the canonical reovirus receptor [31,32,33], which is ubiquitously expressed throughout the body and has several important roles in normal cellular processes including tight junction formation, leukocyte migration, and angiogenesis [34]. Fortuitously, JAM-A is also overexpressed in several cancers, including both haematological and solid malignancies [35,36,37,38,39,40,41]. Following reovirus engagement with JAM-A and receptor-mediated endocytosis, the viral particle undergoes acid-dependent cathepsin-mediated proteolysis within the endosome [42,43] to form an intermediate subviral particle (ISVP) characterised by the loss of σ3 and cleavage of µ1 [44]. The proteolytic uncoating, principally by cathepsins L and B, is critical for penetration of the endosome membrane by µ1; ISVPs undergo a conformational change causing autocleavage of µ1 into µ1N which triggers pore formation in the endocytic membrane [45] and delivers transcriptionally active reovirus into the cytosol [46,47] for replication. Capped, positive-sense single stranded (ss) RNA serves as mRNA for protein translation and provides a template for replication of nascent dsRNA genomes [48]. Transcription and translation occur in cytoplasmic “viral factories” [49,50], with packaging of the segmented genome into virions occurring concomitantly with RNA synthesis [51,52]. Viral egress can be non-cytolytic in the absence of transformation; however, the release of progeny virus is typically lytic in permissive, transformed cells [53,54].

The molecular features associated with the oncolytic capacity of reovirus have been the subject of decades of research. Initially, an association between reovirus permissiveness and epidermal growth factor receptor (EGFR) status was revealed [55,56], along with evidence that activation of downstream signalling pathways, induced after transfection with the oncogene *v-erb*, are important [57]. Subsequent transfection of cells with constitutively active elements of the RAS pathway, a group of small GTP-binding proteins that regulate cell fate and growth, identified a role for RAS in reovirus permissiveness [58]. Therefore, although JAM-A is important for host cell entry, gain-of-function mutations activating RAS signalling [59] could promote reovirus replication and the release of virus progeny [60]. *RAS* mutations are prevalent in cancer [61], supporting the use of reovirus as a potential therapeutic agent [58,62]. The link between reovirus and cellular RAS status was further strengthened by observations that tumour cell susceptibility could be influenced by modulating RAS and/or its downstream effectors using short-hairpin RNA or small-molecule inhibitors [63,64]. Mechanistically, modulation of RAS signalling may promote susceptibility via inhibition of PKR [58]. In healthy cells, binding of dsRNA by PKR results in its dimerization, autophosphorylation and activation. Activated PKR subsequently phosphorylates the translation initiation factor, eIF2, rendering it inactive, which prevents the translation of viral transcripts [65]; however, in *RAS*-transformed cells PKR remains inactive and viral replication can occur [58,66,67]. Currently, the mechanism that coordinates *RAS*-transformation and PKR inactivation remains unclear [68].

Although the RAS–PKR axis provides a plausible explanation for the susceptibility of cancer cells to reovirus, the true molecular mediator has been the subject of debate, with doubt being cast by the survival of some infected *RAS*-transformed cells [69,70]. Moreover, the absence of a correlation between total or phospho-PKR with RAS expression or cell death contradicts previous studies [71], as does the lack of association between oncolysis and EGFR signalling [72]. It has become increasingly apparent that viral replication and cell death are not inextricably linked. Indeed, it is possible that RAS activation does not underlie viral replication but rather sensitivity to apoptosis which can occur independently of replication [53,64]. Sensitivity to reovirus oncolysis is likely to be dependent on multiple cellular and molecular determinants, many of which may yet be undiscovered.

## 4. Mechanisms of Oncolysis

Reovirus was originally considered to operate predominantly by apoptosis (reviewed in [73]). The apoptotic signalling often displayed by infected cells includes the generation of IFN and activation of NF-κB, either through detection of cytoplasmic dsRNA via PKR, retinoic acid-inducible gene I (RIG-I) or melanoma differentiation-associated protein 5 (MDA5), or following σ1 and µ1 receptor engagement or membrane penetration [53,74,75,76,77]. In response to NF-κB and/or IRF3 signalling, inflammatory cytokines such as TNF-related apoptosis-inducing ligand (TRAIL) are secreted, which bind to surface death receptors and trigger activation of caspase-3 and -7 [78,79,80]. While IFN is a potent promoter of cell death, it can be dispensable for reovirus-induced apoptosis, which explains the ability of infected, IFN-deficient tumour cells to undergo apoptosis [79,81]. Blockade of apoptotic caspases does not always abrogate reovirus-induced cell death, indicating that other modes of cell death can also occur [82]. Necroptosis, contingent on recognition of viral dsRNA and induction of a type I IFN response [83], and autophagy following acute endoplasmic reticulum (ER) stress [84] have both been identified as alternative modes of reovirus-induced cell death. Thus, reovirus-induced death is exquisitely linked to the phenotype of the target cell and the surrounding TME; indeed, our recent unpublished data suggest that modulation of pro- vs. anti-apoptotic proteins upon co-culture with stromal cell support can abrogate reovirus-induced apoptosis in malignant B cells. Therefore, examination of viral replication and/or oncolysis in multiple cancer models, and in the context of TME support, will be essential to identify mechanisms of cancer-selective activity and cell death.

## 5. Reovirus Modulation of the Immune System

### 5.1. Reovirus-Induced Innate Anti-Tumour Immunity

Immune cells and infected tumour cells secrete pro-inflammatory cytokines and chemokines in response to reovirus treatment [85,86,87,88]. This occurs via engagement of pathogen-associated molecular patterns (PAMPs; e.g., viral RNA, DNA or proteins) or damage-associated molecular patterns (DAMPs; e.g., heat-shock proteins, calreticulin, uric acid and ATP released from infected cells) with pattern recognition receptors (PRRs) [89]. As with most viral infections, the secretion of type I IFN is a key component of the innate response to reovirus [90]. Viral dsRNA in the cytoplasm of infected cells is detected by PRRs such as RIG-I, MDA5, PKR or Toll-like receptor-3 [91,92] and triggers the transcription of type I IFNs from both infected tumour cells and immune cells; dendritic cells (DCs) and monocytes are important in the detection of reovirus and secretion of IFN-α [35,85,93]. Indeed, specific roles for RIG-I and mitochondrial antiviral-signaling protein (MAVS) but not MDA5 have been reported for reovirus activation of IRF3/IRF7, whilst reovirus activation of an NF-κB was dependent on MDA5 [76]. Moreover, it has been suggested that long reovirus dsRNA gene segments activate MDA5 while short dsRNA segments activate RIG-I [76]. Importantly, a role for RIG-I signaling has also been implicated in reovirus permissiveness of *RAS*-transformed cells; the MEK/ERK pathway—downstream of RAS—blocks signaling from RIG-I and inhibits IFN production, thus enabling reovirus replication [94]. In addition, a role for TLR-3 has been described for reovirus detection within the TME. Here, reovirus inhibited the immunosuppressive activity of myeloid-derived suppressor cells (MDSCs) in a TLR-3-dependent manner [95].

The generation of a pro-inflammatory environment reverses the immunosuppressive state of the TME, induces cytotoxic bystander cytokine killing of tumour cells, activates and recruits innate immune effector cells to kill neoplastic cells, and facilitates the generation of an adaptive anti-tumour immune response [96,97,98,99]. Reciprocal cell-to-cell interactions between DCs and natural killer (NK) cells within the TME or tumour-draining lymph nodes, can stimulate both NK cell activation and DC maturation [85,100]; NK cell anti-tumour immunity within peripheral blood mononuclear cells (PBMCs) is mediated by type I IFN secretion from monocytes [35]. In addition to the recruitment and activation of NK cells, reovirus also activates innate T cells which are capable of eliminating tumour cells via the release of cytolytic granules [85,101]; this remains a poorly understood mechanism of action.

### 5.2. Adaptive Anti-Tumour Immunity

In addition to PAMPs and DAMPs, tumour-associated antigens (TAAs) are also released into the TME during oncolysis. TAAs are phagocytosed by antigen presenting cells (APCs), such as DCs, and the cytokine-rich milieu stimulates DC maturation [102]. Reovirus-activated DCs cross-present TAAs via major histocompatibility complex (MHC) class I to naive CD8^+ve^ T cells [102,103]. These processes facilitate the priming of tumour-specific cytotoxic T lymphocytes (CTLs) [89,102,103]. Interestingly, direct reovirus oncolysis is not essential to generate adaptive anti-tumour immunity, as tumour-specific CTLs have been successfully generated against reovirus-resistant melanoma cells in vivo [104]. Thus, even if a particular cancer is not killed directly by the lytic effects of reovirus, reovirus treatment may offer immunotherapeutic value for patients. By contrast, a recent study by Martin et al., [105] suggested that reovirus was ineffective at priming a systemic immune response compared to alternative OVs, despite effective eradication of the primary tumour. These conflicting data are difficult to interpret; however, the discrepancies observed could be due to the different mouse strains; previous studies [103,104] have utilized T_H_1-dominant C57BL/6 mouse models, whilst this later study used T_H_2 dominant Balb/c mice. Of note, Martin et al., did not examine the induction of tumour specific CTLs but eradication of a secondary tumour. Therefore, it is possible that reovirus did prime effector CTLs which were inhibited due to the upregulation of immune checkpoint molecules, such as programmed death-ligand 1 (PD-L1), or the induction of regulatory T cells (T_regs_) within the TME. Indeed, it is important to note that reovirus can promote the accumulation of T_regs_ and MDSCs [106,107,108] and also upregulates immune checkpoint molecules [108,109,110], which could impede both effector NK cell and CTL responses. Figure 2 (the inner circle) provides an overview of known reovirus mechanisms of action, including oncolysis and the induction of innate and adaptive anti-tumour immunity.

In a recent phase I study of intravenous (i.v.) reovirus there was an increase in transcripts of the pro-recruitment chemokines macrophage inflammatory protein (MIP)-1α and MIP-1β in tumour RNA and in the expression of the intracellular adhesion molecule 1 (ICAM-1) by T cells 48–72 h after infusion [109]. Along with CD68^+ve^ myeloid cells, tumours of reovirus-treated vs. control patients appeared to contain a higher number of CD8^+ve^ T cells [109], whose presence is strongly associated with superior outcomes [111]. Moreover, pro-inflammatory cytokines and IFN were upregulated in the serum of reovirus-treated patients [109,112], which can promote APC maturation and activate NK and T cells, as evidenced by the increased expression of CD69 [113]. Collectively, the evidence suggests that, as an immune adjuvant, reovirus can promote leukocyte infiltration into tumours and support tumour immune surveillance. However, to promote and sustain reovirus-induced anti-tumour immunity it is essential that long-term characterisation of the TME after reovirus treatment is carried out and that combination strategies are developed to counteract any inhibitory/regulatory mechanisms that develop.

### 5.3. The Antiviral Immune Response

The “antiviral” immune response is designed to combat the invading pathogen; however, it could also be fundamental to OV efficacy because of the overlap with “anti-tumour” processes. The humoral arm of adaptive immunity plays an important role in preventing reovirus infection through the generation of neutralising antibodies (NAbs) and there is evidence that circulating reovirus-specific antibodies can impair viral persistence and access to tumours [114]. As reovirus is ubiquitous in the environment [115], the global seroprevalence among adults is commonly above 50% and typically closer to 100% [116,117,118,119,120]. While NAbs may have a positive effect in protecting against reovirus infection, their effect on reovirus therapeutic activity remains controversial. Interesting, but generally less considered in relation to OV therapy, is the fact that viral antigens also prime virus-specific T cells [98,121,122]. These could either potentiate anti-cancer activity through eradication of virally infected tumour cells or abrogate anti-cancer activity by abrogating viral replication and direct oncolysis.

## 6. Reovirus Delivery—Systemic vs. Intra-Tumoural

Although the mechanisms by which reovirus exerts its cytotoxic effects have been the subject of some debate, the fact that it can reliably do so against malignant targets remains unquestioned. Reovirus has oncolytic activity against the vast majority of solid tumour types in vitro (lung, breast, ovarian, prostate, colorectal, pancreatic, glioma, melanoma, and head and neck squamous cell carcinoma (HNSCC)) [72,87,93,123,124,125,126] and has shown promise in haematological models, such as multiple myeloma and both lymphoid and myeloid leukaemias [35,37,127].

When first used as a cancer therapeutic in pre-clinical in vivo models, reovirus was delivered by the intra-tumoural (i.t.) route [128] and induced regression of established subcutaneous B16 melanomas [129], colorectal liver metastases [70] and subcutaneous and orthotopic gliomas [130]. Interestingly, i.t.-administered UV-inactivated reovirus also controlled tumour growth via immune-mediated mechanisms in a liver cancer model [131]. However, the systemic administration of virus into the bloodstream would appear to have the greatest potential to access disseminated tumour cells within the vasculature or distant organs. This is of clinical importance given that metastasis causes ~90% of all cancer-related deaths [132]. Oral intake, by far the most convenient route of systemic drug administration, is not suited to OV therapy as the virus is a gastrointestinal pathogen and is contained within the gastrointestinal system. Vascular injection is therefore the preferred systemic delivery route, being less invasive than locoregional administration. Unfortunately, the impact of i.v. reovirus upon tumour growth is often limited in comparison to i.t. injection; this could be due to: (i) limited delivery to the tumour; (ii) the generation of NAbs resulting in virus neutralisation prior to tumour access; and/or iii) reduced recruitment of immune effector cells to the tumour site.

Because of the size of the typical therapeutic OV infusion (10^9^–10^10^ pfu), B cell mobilisation and antibody production occurs rapidly. From a not-insubstantial baseline, anti-reovirus antibody titres commonly increase ~1000-fold [133] and is greater in response to i.v. than i.t. injection [112]. Strategies to reduce and/or counteract reovirus NAbs have involved the use of immunosuppressive chemotherapy, particularly cyclophosphamide (CPA). CPA can deplete T_regs_ and boost T cell anti-tumour immunity [134]; however, at higher doses, it can suppress the effector functions of all lymphocytes, including B cell antibody production [135,136]. In preclinical models, CPA successfully curtailed B cell responses and enhanced the persistence of reovirus and delivery to tumours [114,137,138]. CPA and other chemotherapy agents have been used successfully alongside i.v. reovirus in clinical trials to reduce NAbs [139,140], with the exception of one phase I trial where CPA did not attenuate anti-viral responses [141].

In patients, reovirus persists in the bloodstream of seropositive individuals in association with immune cells after i.v. infusion and can gain access to the tumour tissue [133,141]. In a reovirus brain trial (EudraCT) 2011-005635-10), reovirus was found in six of nine brain tumours by immunohistochemistry (IHC) and nine of nine tumours by electron microscopy [109] after a single viral infusion. In its predecessor REO-013, reovirus protein was also found in nine of 10 colorectal cancer liver metastases by IHC [133]. Remarkably, in REO-020, it was in patients exhibiting some of the highest NAb titres that reovirus was successfully detected in the tumour [142]. Therefore, it appears that elimination of circulating NAbs is not essential for effective viral delivery. In fact, NAbs may play an important role in controlling toxicity, a phenomenon highlighted in mice with reduced NAbs (due to CPA treatment), and mirrored in B cell-deficient mice, where reovirus replication occurring in the heart and other organs proved lethal [114]. Although not severe, the identification of occasional hepatic and cardiac toxicities in some trials combining reovirus with chemotherapy emphasises the importance of NAbs in systemic virotherapy [140]. Perhaps a more important consideration in this matter is that immunosuppressive agents such as CPA could also dampen cell-mediated immunity [136] and compromise the development of long-term anti-tumour immune responses. Thus, identifying appropriate dosing schedules is essential. For example, low-dose CPA effectively enhances reovirus delivery to tumours while maintaining protective NAb levels [114] and, crucially, has the potential to promote the development of anti-tumour immunity [143,144], although in the context of reovirus this remains unknown

Given the initial belief that NAbs would be detrimental to efficacy, the concept of using cellular chaperones to deliver reovirus to tumours was explored. Immune cells have excellent tumour trafficking potential, and also have the potential to enhance anti-tumour immune effects. When administered i.v., reovirus naturally associates with a number of immune cells in the blood and can be detected on monocytes, NK cells, B cells and granulocytes [109]. Moreover, replication-competent reovirus associates with PBMC in seropositive patients [133,141] and strategies using human PBMC as reovirus carriers have demonstrated that DCs, T cells, and monocytes can act as protective cell carriers with efficient “hand-off” to tumour cells, despite pre-existing antiviral immunity [145,146,147,148]. Similarly, a heterogeneous population of lymphokine-activated killer cells and DCs can deliver reovirus to ovarian cancer cells in the presence of NAbs [149]. Of particular significance is the fact that mice co-treated with reovirus and granulocyte-macrophage colony stimulating factor (GM-CSF) were dependent on NAbs to achieve effective therapy, indicating that NAbs may in fact promote reovirus efficacy [147].

## 7. Unlocking the Potential of Reovirus with Combination Therapeutics

No matter which route of delivery is chosen, it remains clear that combination therapies will be necessary to optimise reovirus efficacy. Combination with radiotherapy has been investigated on the basis that activating mutations in *RAS* are associated with resistance to radiotherapy but confer sensitivity to reovirus. Twigger et al. reported that this treatment combination increased cell death in a number of cancer cell lines in vitro and in vivo, particularly in cell lines that showed only moderate reovirus sensitivity [150]. Similarly, the combination of reovirus with radiotherapy enhanced therapeutic outcomes in two models of paediatric sarcoma [151]. In both studies, the enhanced therapeutic outcome appeared to be due to increased direct cytotoxicity.

Multiple studies have investigated the combination of reovirus with chemotherapeutic agents, with synergy being frequently observed. As with radiotherapy, the enhanced treatment effect appeared to be due to increased oncolysis. For example, treatment of a range of prostate cancer cell lines with reovirus plus docetaxel, paclitaxel, vincristine, cisplatin or doxorubicin led to increased apoptosis/necrosis in vitro and reovirus improved docetaxel therapy in a xenograft prostate cancer model [152]. Increased apoptosis and/or necrosis has also been demonstrated by the combination of reovirus with: cisplatin in a melanoma model [153]; cisplatin, gemcitabine or vinblastine in non-small cell lung cancer cell lines [125]; and cisplatin plus paclitaxel in both in vitro and in vivo models of head and neck cancer [154]. Collectively, this evidence suggests that the beneficial outcomes resulting from combining reovirus with chemotherapy agents are generally mediated through oncolysis rather than immune-mediated mechanisms. However, Gujar et al. suggested that improved survival following reovirus plus gemcitabine treatment in an ovarian cancer model was at least partly immune-mediated, with reduced numbers of MDSC in tumours and improved anti-tumour CTL responses [155].

The majority of chemotherapeutic agents induce apoptosis, though the mechanisms by which they do this differ: Paclitaxel utilizes different apoptotic pathways depending on its concentration [156]; tamoxifen and gemcitabine activate mitogen-activated protein kinase (MAPK) and p53-dependent pathways and upregulate pro-apoptotic factors [157,158]; while docetaxel induces a non-apoptotic mode of death [159]. Reovirus itself induces apoptosis but can also induce necroptosis, which requires later stages of infection [83]. The reported synergy between reovirus and chemotherapy agents may be due to the induction of this additional form of cell death; however, it could also be due to the ability of reovirus to increase the expression of pro-apoptotic Bcl-2 family proteins [160]. Of particular significance is the dependence of reovirus on apoptosis, which may make it sensitive to resistance mechanisms utilized by cancer cells to escape chemotherapy cytotoxicity. Indeed, our studies have shown that stromal cell support of malignant B cells and multiple myeloma cells can inhibit reovirus sensitivity, in line with that observed for standard of care (SOC) chemotherapy agents (data not shown).

More recently, reovirus has been combined successfully with more targeted cancer therapies. The majority of malignant melanomas carry activating mutations in the RAS-RAF-MEK-ERK signalling pathway, with *NRAS* and *BRAF* mutations being most common. Although inhibition of this pathway might be expected to antagonize reovirus-induced cytotoxicity, the combination of reovirus with small molecule inhibitors of BRAF or MEK actually enhanced ER stress-induced apoptosis [161]. Similarly, the combination of reovirus with bortezomib, a proteasome inhibitor that increases ER stress, increased apoptosis in multiple myeloma cell lines in vitro and improved outcomes in vivo [160]. Energy metabolism within cancer cells is now emerging as an important element for OV susceptibility. OVs, such as reovirus, utilise host metabolic pathways to provide essential nucleotides, lipids, and amino acids for virus propagation and as such, metabolic reprogramming has been considered as a strategy to potentiate OV efficacy [162]. In the context of reovirus, susceptibility has been reported to correlate with pyruvate metabolism and oxidative stress, with a central role for pyruvate dehydrogenase (PDH). Specifically, the early oxidative stress response following reovirus treatment inhibits pyruvate dehydrogenase (PDH), via PDH kinase (PDK) phosphorylation, and induces a metabolic state that does not support reovirus replication. However, reactivation of PDH, using the PDK inhibitors dichloroacetate and AZD7545 enhanced reovirus efficacy in vitro and in vivo. Therefore, metabolic reprogramming is a promising approach to increase the therapeutic potential of reovirus in cancer patients [163]. Another interesting study found that pre-conditioning tumours with bevacizumab—a vascular endothelial growth factor (VEGF) inhibitor—and then withdrawing treatment, rendered endothelial cells susceptible to reovirus infection, induced vascular collapse and promoted immune-mediated tumour clearance [164]. Similar effects were also observed following withdrawal of paclitaxel-mediated inhibition of VEGF signalling [165].

Other combination strategies have focused on boosting immune-mediated anti-tumour effects. For instance, combining reovirus with oncolytic vesicular stomatitis virus (VSV) in a dual-OV “prime-boost” regimen led to improved melanoma therapy via induction of different arms of the immune response; VSV induced a melanoma-specific T_H_17 response which augmented the T_H_1 response induced by reovirus [166]. As discussed above, cell carriage of reovirus by circulating myeloid cells has been potentiated by pre-conditioning the host with GM-CSF to expand immune effector populations [147]. Another strategy that has demonstrated successful results in several cancer models is the combination of reovirus with immune checkpoint inhibitors. Rajani et al. showed that the combination of i.t. reovirus with systemic anti-programmed cell death protein 1 (PD-1) enhanced survival in melanoma-bearing mice compared to either therapy alone [106]. Addition of checkpoint blockade to the dual OV “prime-boost” approach described above also enhanced survival [166]. Three studies have also demonstrated that reovirus can “sensitize” tumours to subsequent checkpoint blockade: (i) reovirus treatment of multiple myeloma cells in vitro increased PD-L1 expression, with systemic reovirus treatment followed by anti-programmed death-ligand 1 (PD-L1) increasing survival in a syngeneic model of multiple myeloma [110]; (ii) increased PD-L1 expression was observed in high grade glioma patients following reovirus treatment and systemic reovirus/anti-PD-1 therapy improved survival in a syngeneic, orthotopic murine glioma model [109]; and (iii) i.t. reovirus increased both PD-L1 expression on tumour cells and the number of intra-tumoral T_regs_ in a murine breast cancer model, while combination reovirus/anti-PD-1 treatment enhanced survival by reducing T_reg_ numbers and improving tumour-specific CTL responses [167]. More recently, reovirus has also been used in combination with CD3-bispecific antibodies. Reovirus-induced IFN stimulated the recruitment of NK cells and reovirus-specific CD8^+^ T cells to the tumour site. Non-exhausted reovirus-specific effector T cells acted in synergy with CD3- bispecific antibodies to reduce the in vivo growth of multiple tumour types including pancreas, melanoma and breast; moreover, reovirus preconditioning was required for maximal efficacy. Importantly, combination treatment was also effective at distant lesions, not injected with reovirus, demonstrating the potential of this strategy for the treatment of metastatic disease.

## 8. Reovirus Clinical Trials

Reovirus T3D is the subject of one of the largest clinical trial programmes in oncolytic virotherapy (OVT). The clinical grade formulation of reovirus is now marketed as pelareorep (formerly Reolysin^®^) by Oncolytics Biotech Inc. (Calgary, AB, Canada). The virus is listed in 26 trials identified on www.clinicaltrials.gov. As of 2018, reovirus holds orphan drug status from the FDA for glioma, ovarian, pancreatic, peritoneal and gastric cancers, and from the European Medicines Agency (EMA, Amsterdam, The Netherlands) for ovarian and pancreatic cancer.

The first-in-man phase I study of reovirus, REO-001, enrolled 19 patients with accessible, advanced malignancies, who were treated intra-tumourally with ascending doses of the virus. No dose-limiting toxicities were observed, all being grade two or below, with nausea, headache or vomiting being the most common [168]. Tumour responses were apparent in 37% of patients. Based on this and its promising safety profile in animal models, reovirus progressed quickly into trials of systemic treatment. Intravenous delivery was first tested in REO-004. Eighteen patients with advanced solid tumours received virus doses of up to 3 × 10^10^ TCID_50_ without identifying dose-limiting toxicity. In fact, only two patients experienced grade two events, even when multiple doses were given on successive days [169]. When corroborated by other phase I trials [112,170], these results demonstrated that when delivered by infusion as a very large, non-physiological bolus, reovirus is remarkably well tolerated. Interestingly, i.v. administration of reovirus in a phase I trial of heavily pre-treated patients with advanced cancers increased the number of CD4^+ve^ T cells, CD8^+ve^ T cells and NK cells, as well as cytokine levels, in the blood, suggesting the onset of an immune response. Significantly, i.v. administration of reovirus in brain tumours also led to a local IFN response with recruitment of CTLs [109].

Reovirus has now undergone further evaluation in phase I and II clinical trials across a range of indications; summarised in Table 1. Historically, the tumours most heavily targeted within the reovirus programme have been melanoma, myeloma and glioma [142,171,172], although trials have also included pancreatic, lung, breast, colorectal, prostate, and head and neck cancers [108,109,173,174,175,176]. Initial trials deployed reovirus as a monotherapy, the majority utilising i.v. administration; safety was established in the almost total absence of serious adverse events [177], with equivocal outcomes reported in phase II trials [142,178]. The mixed outcomes of patient response in clinical trials have made the therapeutic potential of reovirus a topic of debate. It is accurate to state that i.v. reovirus has often shown very modest activity, particularly as a monotherapy [142]. However, it reliably gains access to tumour lesions when administered systemically [109,133]. Currently, the virus is no longer under active investigation as a monotherapy and Oncolytics Biotech Inc. is instead developing combination programmes (www.oncolyticsbiotech.com).

## 9. The Future for Reovirus—Pre-Clinical Requirements and Clinical Considerations

In spite of its efficacy in pre-clinical models, reovirus treatment (as with other OVs) has benefited only a minority of patients. Figure 2 highlights some possible reasons for this and summarizes what we currently know about reovirus (the inner circle) along with some priority areas of research which should aid the development of more effective reovirus therapies (the outer circle). Currently it remains unclear how best to administer reovirus in order to obtain optimal therapeutic responses while maintaining safety. The route designed to maximize efficacy via oncolysis may differ from that designed to facilitate immune-mediated tumour clearance. Although translational studies reliably demonstrate that reovirus can access tumours after i.v. administration [109,133], a greater understanding of the effect of anti-reovirus immunity, both humoral and cell-mediated, is pivotal to maximize its clinical efficacy.

Born of the desire to accelerate clinical application, reovirus has generally been combined with SOC therapies. This has generally led to improved efficacy due to increased cytotoxicity but a more strategic approach, based on a complete understanding of the mechanisms of death induced by each therapy and the challenges faced within defined TMEs, would generate further improvements.

An important aspect of combination therapies is the dosing regimen employed. How many reovirus administrations are required? How frequent should they be? Should they be administered before, after or simultaneously with other agents? Currently, the treatment regimens employed in clinical trials reveal no consensus on what the optimum dosing schedule might be. The planned regimen for the most recent trial is 4.5 × 10^10^ TCID_50_ reovirus i.v. on days 1/2/8/9/15/16 of a 28-day cycle, but other regimens have been used including delivery on days 1/2/3/4/5 of a 28-day cycle or days 1/2/3/8 for the first 21-day cycle and days 1/8 thereafter. These regimens may be pragmatic to facilitate combination with SOC therapies but they may not be the most efficacious. Going forward, it will be important to optimise chemotherapy-induced cytotoxicity while maintaining reovirus-mediated anti-tumour immunity. For example, chemotherapy agents that induce lymphopenia might abrogate immune responses, therefore careful selection of complementary chemotherapies is essential. Indeed, combination of reovirus with gemcitabine can improve anti-tumour immune responses [155] indicating that the two mechanisms can be compatible. Consideration of treatment regimens will be particularly important for combination with immune checkpoint inhibitors because anti-cytotoxic T lymphocyte-associated 4 (CTLA-4) antibodies are likely to potentiate early stages of T cell priming, whilst anti-PD1/anti-PD-L1 antibodies would act to reverse T cell exhaustion within the TME.

Whilst murine pre-clinical models will be essential to identify and validate novel reovirus combinations with improved efficacy, it is important to recognise, and reflect on, the limitations of many commonly used in vivo models. In particular, xenograft models utilizing immunocompromised mice do not consider OV-induced anti-tumour immune responses; moreover, syngeneic tumour models, in immunocompetent mice, do not always model tumour progression at the correct anatomical site. Although more advanced in vivo modes are available (e.g., spontaneous cancer models), which more accurately reflect disease progression, these are expensive and time consuming, restricting their use for many cancer researchers. Importantly, these models do not represent the heterogenous nature of patient tumours. Therefore, it is it imperative that clinical trials are designed to gain as much information as possible. Specifically, clinical trials should allow downstream interrogation of the tumour and the TME, including cancer-associated fibroblasts, immune cell components and soluble factors/extracellular vesicles. Ideally, multiple patient samples (e.g., blood and primary/secondary tumour tissue) should be obtained pre- and post-treatment to gain insight into why some patients may respond, whilst others do not. Detailed characterization of these samples will facilitate the development of more complex combination regimes to counteract resistance mechanisms and allow predictive biomarkers of response to be identified.

While genetic modification of other OV has improved efficacy in pre-clinical models, this approach has not been widely used with reovirus because the segmented RNA backbone makes it difficult to modify. Nevertheless, recent identification and characterisation of reovirus mutants isolated from human U118MG glioblastoma cells has revealed the capacity of JAM-A-independent (jin) mutants to infect JAM-A^−ve^ cells, which are usually resistant to wild-type virus [189]. Following this, a reverse genetics approach was developed to allow genetic modification of expanded-tropism jin mutants [190] and small transgenes including reporter constructs have been inserted [191,192]. This yields tremendous scope to develop novel, genetically engineered reovirus platforms, with enhanced tropism, increased infectivity and replication, and improved immune stimulation. Indeed, reovirus has recently been armed with functional GM-CSF to boost anti-tumour immunity [193]. In addition to the reovirus jin mutants, reassorted reovirus platforms are also undergoing pre-clinical development. Co-infection and serial passage of MDA-MB-231 cells with the prototype laboratory strains for reovirus (type one *Lang*, type two *Jones*, and type three *Dearing*) generated a reassorted virus with a predominant type one genetic composition and some type three gene segments which displayed enhanced infectivity and cytotoxicity in triple-negative breast cancer cells [194]. Moreover, the advancement of reovirus engineering has enabled mutations to be made that can counteract inhibitory mechanisms within the TME. In particular, mutations within σ1 have been incorporated to prevent proteolytic cleavage of σ1 by breast cancer-associated proteases, which abrogated binding to sialic acid; infectivity was restored in the σ1 mutants [195]. These innovations suggest a new and exciting era of reovirus research is emerging.

No single OV has emerged as the undisputed leader in terms of efficacy and it is unlikely that a “one size fits all” OV exists. Having demonstrated some clinical activity, reovirus remains a promising weapon in the cancer therapy arsenal where viral modifications, allied with informed scheduling and strategic combination with other treatments, should pay dividends for cancer patients.

## Figures and Tables

**Figure 1 cancers-12-03219-f001:**
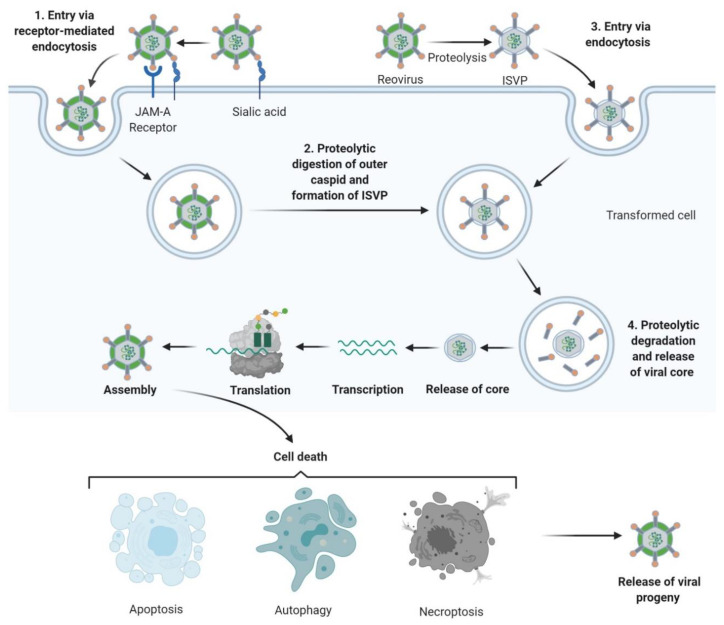
Reovirus replication: **1**. Reovirus is first tethered via a weak interaction between σ1 and cell surface sialic acid; σ1 then binds with high affinity to junctional adhesion molecule A (JAM-A) resulting in internalization of the virus via receptor-mediated endocytosis. **2.** Once internalized, the virus is transported to early and late endosomes where it undergoes proteolytic digestion to remove the outer capsid protein σ3 resulting in the formation of infectious subvirion particles (ISVPs). **3.** Alternatively, ISVPs may be formed by extracellular proteases within the tumour environment allowing direct entry into cells via membrane penetration. **4.** After further proteolytic degradation a transcriptionally active viral core is released into the cytoplasm. Transcription and translation occur ultimately leading to the assembly of new viral progeny, host cell death and progeny release. Figure created using Biorender (https://biorender.com/).

**Figure 2 cancers-12-03219-f002:**
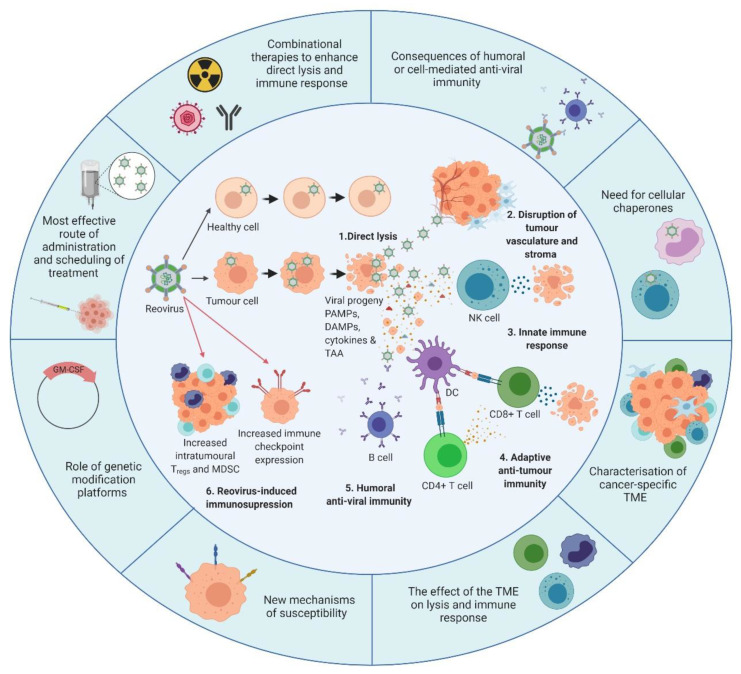
Overview of reovirus mechanisms of action and the developments required. The **inner circle** illustrates what is currently known about reovirus. **1.** In healthy cells, anti-viral immune responses limit reovirus replication and prevent lytic killing. By contrast, oncogenic signalling pathways render tumour cells susceptible to reovirus replication and direct oncolysis. **2.** Reovirus replicates in the tumour vasculature and stroma due to reciprocal cell:cell interactions which alter anti-viral signalling. **3.** Infection of tumour cells leads to the release of viral progeny, cytokines and tumour-associated antigens (TAAs), which initiates innate anti-tumour immunity including cytokine-mediated bystander killing and natural killer (NK) cell-mediated cytotoxicity. **4.** Adaptive anti-tumour immunity is generated following the phagocytosis of TAAs by dendritic cells (DCs) and presentation of TAAs to CD4^+ve^ and CD8^+ve^ T cells, which facilitates priming of tumour-specific cytotoxic T lymphocytes (CTLs). **5.** In addition to innate and adaptive anti-tumour immune responses, humoral anti-viral immunity is induced, leading to the production of reovirus-specific neutralising antibodies (NAbs). **6.** Following induction of anti-tumour/anti-viral immune responses, regulatory immune mechanisms are “switched-on” to control ongoing immune responses, including upregulation of immune checkpoints and increased levels of regulatory T cells (T_regs_) and/or myeloid-derived suppressor cells (MDSCs). The **outer circle** highlights priority research areas to improve reovirus efficacy. These include gaining a greater understanding of: (i) the consequence of humoral and/or cell-mediated anti-viral immunity on reovirus efficacy which would inform the development of, or the requirement for, cellular chaperones; (ii) the tumour microenvironment (TME) and how it influences reovirus oncolysis and anti-tumour immunity; (iii) the cellular determinants utilized by reovirus for direct oncolysis, including mechanisms of reovirus resistance; (iv) the potential benefits of genetically-modified reovirus platforms; (v) reovirus scheduling to maximize virus delivery and efficacy including the best route of virus administration; and (vi) combinatorial approaches that are designed to boost both direct oncolysis and anti-tumour immune responses. PAMPs: pathogen-associated molecular patterns; DAMPs: damage-associated molecular patterns; GM-CSF: granulocyte-macrophage colony stimulating factor. Figure created using Biorender.com.

**Table 1 cancers-12-03219-t001:** Summary of Reovirus Clinical Trials.

Disease	Combinations	Phase	Trial ID	Route	Dose(s) TCID_50_	Results
Gliomas	N/A	I	NCT00528684	I.T	1 × 10^7^, 1 × 10^8^, 1 × 10^9^	No DLT, 10/12 patients had PD, 1/12 SD and 1/12 patients unevaluable for response, but alive >4.5 years post treatment [171].
	N/A	I	EudraCT 2011-005635-10	I.V	1 × 10^10^	Reovirus detected in within tumours and increased CTL infiltration [109].
Brain cancer	Sargramostim (GM-CSF)	I	NCT02444546	I.V	MTD	Ongoing
Pancreatic cancer	Carboplatin and Paclitaxel	II	NCT01280058	I.V	3 × 10^10^	No significant enhancement of PFS with reovirus combination therapy (*n* = 36) vs. Carboplatin/Paclitaxel alone (*n* = 37) (4.9 vs. 5.2 months) [108].
	Pembrolizumab and 5 Fluorouracil or gemcitabine or irinotecan	I	NCT02620423	I.V	4.5 × 10^10^	Well tolerated. 3/10 evaluable patients had SD, 1 of which had PR for 17.4 months. Biopsies show reovirus infection in tumour cells and immune infiltrates [179].
	Pembrolizumab	II	NCT03723915	I.V	Not reported	Ongoing
	Gemcitabine	II	NCT00998322	I.V	1 × 10^10^	Well tolerated. 1/29 patients had PR, 23/29 SD, 5/29 PD. Single patient with SD had upregulated expression of PD-L1 following treatment [180].
Colorectal cancer	Irinotecan and Leucovorin and 5-Fluorouracil	I	NCT01274624	I.V	1 × 10^10^– 3 × 10^10^	2/21 patients had DLT. 18/21 evaluable for response. 1/18 PR, 9/18 SD, 8/18 PD [174].
	Leucovorin and 5-Fluorouracil and Oxaliplatin and Bevacizumab	II	NCT01622543	I.V	3 × 10^10^	Poorer PFS with reovirus combination therapy (7 months vs. 9 months). No significant difference in OS [181].
Head and Neck Cancers	Carboplatin and Paclitaxel	II	NCT00753038	I.V	3 × 10^10^	Well tolerated. 4/13 evaluable patients had PR, 2/13 had SD for >12 weeks [173].
	Carboplatin and Paclitaxel	III	NCT01166542	I.V	3 × 10^10^	Interim results reported (www.oncolyticsbiotech.com). 118 evaluable patients, reovirus increased PFS from 48 to 95 days. Significantly increased OS. Curtailed to larger phase II trial.
Melanoma	N/A	II	NCT00651157	I.V	3 × 10^10^	Well tolerated, viral replication was detected in 2/15 patients despite NAb, average PFS 45 days [142].
	Carboplatin and Paclitaxel	II	NCT00984464	I.V	3 × 10^10^	Well tolerated. 3/14 patients had PR, 9/14 SD, 2/14 PD. ORR of 21%, no complete responses [182].
Multiple Myeloma	N/A	I	NCT01533194	I.V	3 × 10^9^, 3 × 10^10^	No DLT reported, reovirus localization to BM, SD for up to 8 months [172].
	Lenalidomide or Pomalidomide	I	NCT03015922	I.V	3 × 10^10^	Ongoing
	Dexamethasone and Carfilzomib	I	NCT02101944	I.V	MTD	Recruiting
	Dexamethasone and Bortezomib	I	NCT02514382	I.V	MTD up to 4.5 × 10^10^	Ongoing
	Dexamethasone and Carfilzomib and Nivolumab	I	NCT03605719	I.V	MTD	Recruiting
Lung Cancer	Carboplatin or Paclitaxel	II	NCT00861627	I.V	3 × 10^10^	11/37 of patients PR, 20/37 SD, PFS 4 months [175].
		II	NCT00998192	I.V	3 × 10^10^	Treatment well tolerated, 12/25 patients had PR, 10/25 SD, 3/25 PD [183].
	Pemetrexed or Docetaxel	II	NCT01708993	I.V	4.5 × 10^10^	Virus was well tolerated, no enhancement of PFS with reovirus vs. drugs alone (2.96 vs. 2.83 months) [184].
Prostate cancer	Docetaxel and Prednisone	II	NCT01619813	I.V	3 × 10^10^	Poorer OS in virus and drug combination arm, vs. drug alone [185].
Breast cancer	Paclitaxel	II	NCT01656538	I.V	3 × 10^10^	Combination arm showed improved OS vs. drug alone arm (17.4 vs. 10.4 months) [176].
	Avelumab and Paclitaxel	II	NCT04215146	I.V	4.5 × 10^10^	Recruiting
	Retifanlimab	II	NCT04445844	I.V	MTD	Recruiting
Ovarian cancer	Paclitaxel	II	NCT01166542	I.V	3 × 10^10^	Median PFS 4.3 months and ORR 20% for patients receiving Pacitaxel alone vs. 4.4 months and 17.4%, for combination treatment. Addition of reovirus to treatment does not reduce the hazard of progression or death [186].
Bone and soft tissue sarcoma	N/A	II	NCT00503295	I.V	3 × 10^10^	Well tolerated. 14/33 patients had SD for >2 months, including 5 patients which had SD for >6 months [178].
Advanced cancer	Radiotherapy	I		I.T	1 × 10^8^–1 × 10^10^	No DLT. Low dose radiation arm 2/7 PR and 5/7 SD. High dose radiation arm 5/7 PR and 2/7 SD [187].
	Carboplatin and Paclitaxel	I		I.V	3 × 10^9^, 1 × 10^10^, 3 × 10^10^	No DLT. 1/26 patients had CR, 6/26 PR, 9/26 SD, 2/25 major clinical response, and 9/25 PD [139].
	Docetaxel	I		I.V	3 × 10^9^, 1 × 10^10^, 3 × 10^10^	MTD not reached. 1/16 patients had CR, 3/16 PR, 3/16 minor response, 7/16 SD, 2/16 PD [188].
	Gemcitabine	I		I.V	1 × 10^9^ 3 × 10^9^, 1 × 10^10^, 3 × 10^10^	3/16 patients had DLT. 10/16 patients evaluable for response, 1/10 PR, 6/10 SD, 3/10 PD [140].

DLT: dose-limiting toxicity, PFS: progression-free survival, PR: partial response, ORR: overall response rate, SD: stable disease, PD: progressive disease, IT: intra-tumoural, I.V: intravenous, MTD: maximum tolerated dose, PD-L1: anti-programmed death-ligand 1, BM: bone marrow, OS: overall survival.

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
