# Peer review of "Past, Present and Future of Oncolytic Reovirus"

_cancers, 2020, doi:10.3390/cancers12113219_

Round 1

Reviewer 1 Report

An excellent historical review of reovirus as a potential cancer therapeutic. Well written and a nice review of reoviral biology  and clinical trial activity to date.

This review would be strengthened by a more in depth outline of the future strategies for clinical reovirus trials including a more in depth analysis of host  and tumour factors predicting OV therapeutic efficacy and synergies with other cancer therapeutics.

Author Response

Reviewer 1 suggestion: This review would be strengthened by a more in depth outline of the future strategies for clinical reovirus trials including a more in depth analysis of host and tumour factors predicting OV therapeutic efficacy and synergies with other cancer therapeutics.

We would like to thank reviewer one for this suggestion and have added an additional paragraph on page 19, lines 562-578, to discuss this issue. 

Reviewer 2 Report

In this review, Müller et al provide an overview of oncolytic reovirus, detailing their mechanisms of action and anti-tumor features, pre-clinical and clinical studies, and discussing the potential of synergistically combining reovirus together with other therapeutics. The review is interesting and comprehensive on this topic.

Due to the growing interest on targeted metabolic reprogramming of TME as a combinatorial strategy to enhance OVs anti-tumor function, a comment on recent findings would be greatly appreciated (as an example: DOI: 10.1158/0008-5472.CAN-18-2414).

Minor revision:

  • Please edit first paragraph: rows 38 to 42 are not useful to highlight OV potential, it is suggested to eliminate or rephrase.
  • The reference to ECHO-7 is unnecessary due to the revocation of the license for clinical use, as authors specify.

Author Response

Reviewer 2 suggestions: Due to the growing interest on targeted metabolic reprogramming of TME as a combinatorial strategy to enhance OVs anti-tumor function, a comment on recent findings would be greatly appreciated (as an example: DOI: 10.1158/0008-5472.CAN-18-2414)

We agree, this is an important research area that was omitted in the original article.  To address this deficiency, we have added a new paragraph; page 11-12 lines 379=389.

Suggested minor revisions:

  1. Please edit first paragraph: rows 38 to 42 are not useful to highlight OV potential, it is suggested to eliminate or rephrase.

This historical introduction has now been deleted; see page 2, lines 67-74.

  1. The reference to ECHO-7 is unnecessary due to the revocation of the license for clinical use, as authors specify.

The reference to ECHO-7 has now been deleted; see Page 2, lines 71-74.  

Reviewer 3 Report

Important and highly informative review, covering many different aspects of reovirus as OV, from biology to clinical application. Remaining topics of debate and shortcomings in current knowledge are highlighted in an unbiased way. The review is very well written, it was a pleasure to read.

Still some sections need further clarification:

  1. Page 2 lines 74-78. In general, reovirus has been considered safe, however, sporadic reports of severe pathology and association with coeliac disease were mentioned. Are there any implications from these studies for its use as anti-cancer agent?
  2. Page 3, line 122. Please explain how PKR is phosphorylated in normal cells and how this is different in ras transformed cells.
  3. Page 5, section on innate anti-tumour immunity: Line 155, subtitle is cryptic. Please rephrase
  4. Lines 157-162. In this section, the authors discuss how reovirus induce innate immunity however, it remains quite general. It is not clear which PAMPS and DAMPs are specific for reovirus/ reovirus-infected cells and which PRRs are specifically involved in recognition of reovirus. The cited papers discuss sensing of RNA viruses in general, but do not specifically address reovirus. Similarly, in lines 137 – 139 on the sensors of reovirus in the context of oncolysis, refs are missing. Is the mechanism of reovirus recognition still an open issue? To my knowledge at least 1 paper has described PRRs involved in sensing of reovirus. Authors may consider to cite this paper: Goubau et al, Nature 2014 (DOI: 1038/nature13590). If PRR for reovirus are still partly unresolved, this needs to be mentioned.
  5. Page 175-189, section on adaptive anti-tumor immunity. Lines 178-179. Ref is missing.
  6. In the same section, the authors discuss several papers demonstrating that reovirus can promote anti-tumor adaptive immunity, however, the authors did not discuss the recent work of Martin et al in Sci Reports 2019 (DOI: 1038/s41598-018-38385-7). In this paper, the authors show that reovirus does not induce anti-tumor immunity in contrast to other OVs. These conflicting results need to be discussed.
  7. Page 8 lines 239-249. It is unclear whether CPA treatment only affects the B-cell/antibody responses or also attenuate T cell mediated cellular immune responses to reovirus. Is this known?
  8. Page 13, lines 433-444 may fit better in the section on combination treatment (lines 281).
  9. Authors may consider to include the recently published paper by Groeneveldt et al, JITC 2020 (doi: 10.1136/jitc-2020-001191) on combination of reovirus and bi-specific antibodies. Here, also the timing of different therapies is investigated, as discussed in lines 446-447.
  10. The section on clinical trials is quite long and descriptive. Authors may consider to summarize (part) of the information in a table.
  11. Figures are nice and informative but seem to have a low resolution.

Author Response

Reviewer 3 suggestions: Sections for further clarification:

  1. Page 2 lines 74-78. In general, reovirus has been considered safe, however, sporadic reports of severe pathology and association with coeliac disease were mentioned. Are there any implications from these studies for its use as anti-cancer agent?

In response to this suggestion we have expanded this statement on page 3, lines 95-96.   

  1. Page 3, line 122. Please explain how PKR is phosphorylated in normal cells and how this is different in ras transformed cells.

Additional details regarding PKR phosphorylation have been added on page 5, line 148.

  1. Page 5, section on innate anti-tumour immunity: Line 155, subtitle is cryptic. Please rephrase

This has been changed to “reovirus-induced innate anti-tumour immunity”. See page 6, line 181.

  1. Lines 157-162. In this section, the authors discuss how reovirus induce innate immunity however, it remains quite general. It is not clear which PAMPS and DAMPs are specific for reovirus/ reovirus-infected cells and which PRRs are specifically involved in recognition of reovirus. The cited papers discuss sensing of RNA viruses in general, but do not specifically address reovirus. Similarly, in lines 137 – 139 on the sensors of reovirus in the context of oncolysis, refs are missing. Is the mechanism of reovirus recognition still an open issue? To my knowledge at least 1 paper has described PRRs involved in sensing of reovirus. Authors may consider to cite this paper: Goubau et al, Nature 2014 (DOI: 1038/nature13590). If PRR for reovirus are still partly unresolved, this needs to be mentioned.

In response to this suggestion we have added a paragraph specifically related to the detection of reovirus, which includes additional references; see page 6 lines 190-199.  We have also incorporated additional references for the earlier section where these were missing (see page 6, line 165).

  1. Page 175-189, section on adaptive anti-tumor immunity. Lines 178-179. Ref is missing.

References for this have now been added (page 6, line 217); however, reference to CD4 T cells has been deleted as this has not been defined specifically for reovirus but is rather inferred because of the requirement for CD4 T cell help for the priming of cytotoxic T lymphocytes.

  1. In the same section, the authors discuss several papers demonstrating that reovirus can promote anti-tumor adaptive immunity, however, the authors did not discuss the recent work of Martin et al in Sci Reports 2019 (DOI: 1038/s41598-018-38385-7). In this paper, the authors show that reovirus does not induce anti-tumor immunity in contrast to other OVs. These conflicting results need to be discussed.

Thank you for pointing out this paper and the conflicting results reported.  Some discussion of this paper and the contradictory data presented has been added to page 7, lines 221-230.

  1. Page 8 lines 239-249. It is unclear whether CPA treatment only affects the B-cell/antibody responses or also attenuate T cell mediated cellular immune responses to reovirus. Is this known?

I have added details to state that low dose CPA can boost T cell responses via decreased T regs (page 8, line 293); however, I was unable to find reovirus specific papers that have examined the induction of adaptive immunity in the presence of CPA as most papers have examined reovirus delivery.  A statement has also been added to page 10, line 318 to clarify that the ability of CPA to enhance reovirus-induced T cell immune responses remains unknown.

  1. Page 13, lines 433-444 may fit better in the section on combination treatment (lines 281).

We agree this section is better placed in the combination therapy section and have move this accordingly. It is now located on page 11, lines 359-370, and has been deleted from page 18.

  1. Authors may consider to include the recently published paper by Groeneveldt et al, JITC 2020 (doi: 10.1136/jitc-2020-001191) on combination of reovirus and bi-specific antibodies. Here, also the timing of different therapies is investigated, as discussed in lines 446-447.

Thank you for highlighting this recent publication which should certainly be included in the review.  We have added details of this publication and the combination strategy on page 12, lines 414-421. 

  1. The section on clinical trials is quite long and descriptive. Authors may consider to summarize (part) of the information in a table.

Much of this section has now been summarised in an additional table – please see Table 1 starting on page14.  In accordance with this, text within this section has also been deleted, see page 13 onwards.

  1. Figures are nice and informative but seem to have a low resolution.

We have re-inputted the figures from JPEG files to increase the resolution and enlarged figure 1 (page 4) to make it more visible.  We hope this has helped with the resolution of these figures.  

Reviewer 4 Report

Additional experimental work is necessary before this manuscript could be considered for publication in this Journal. As well, a revised manuscript should address inconsistencies between the presented work and published literature.

Comments

  1. Lines 281 to 336 should be summarized in a figure or table.
  2. Lines 337 to 416 should be summarized in a table.

Author Response

Reviewer 4 suggestions: Additional experimental work is necessary before this manuscript could be considered for publication in this Journal. As well, a revised manuscript should address inconsistencies between the presented work and published literature.

Reviewer 4 has indicated that further experimental work is required prior to manuscript publication.  However, as the article is review of the current literature, we do not believe it should contain additional experimental data, thus, we do not think this comment is applicable.  

  1. Lines 281 to 336 should be summarized in a figure or table.

Given the multiple combination strategies summarised within this section and the limited time given for the revisions we have not been able to generate this additional figure to summarise this section.

  1. Lines 337 to 416 should be summarized in a table.

In response to this suggestion, we have now summarised much of this information in an additional table – please see table 1 on page 14 onwards.